# Sustained Activation of Myeloperoxidase Is Associated with Oxidative Stress and Inflammation in People Living with the Human Immunodeficiency Virus at Risk of Cardiovascular Disease

**DOI:** 10.3390/ijms262110285

**Published:** 2025-10-22

**Authors:** Haskly Mokoena, Joel Choshi, Sidney Hanser, Sihle E. Mabhida, Helen C. Steel, Kabelo Mokgalaboni, Wendy N. Phoswa, Gerald Maarman, Bongani B. Nkambule, Phiwayinkosi V. Dludla

**Affiliations:** 1Department of Physiology and Environmental Health, University of Limpopo, Sovenga 0727, South Africa; haskly.mokoena@ul.ac.za (H.M.); joel.choshi@ul.ac.za (J.C.); 2Department of Physiology, School of Medicine, Sefako Makgatho Health Science University, Ga-Rankuwa, Pretoria 0208, South Africa; 3Non-Communicable Diseases Research Unit, South African Medical Research Council, Tygerberg 7505, South Africa; sihle.mabhida@mrc.ac.za; 4Department of Immunology, University of Pretoria, Pretoria 0001, South Africa; helen.steel@up.ac.za; 5Department of Life and Consumer Sciences, College of Agriculture and Environmental Sciences, University of South Africa, Florida Campus, Roodepoort 1710, South Africa; mokgak@unisa.ac.za (K.M.); phoswwn@unisa.ac.za (W.N.P.); 6CARMA: Centre for Cardio-Metabolic Research in Africa, Division of Medical Physiology, Department of Biomedical Sciences, Faculty of Medicine & Health Sciences, Stellenbosch University, Cape Town 8000, South Africa; gmaarman@sun.ac.za; 7School of Laboratory Medicine and Medical Sciences, University of KwaZulu-Natal, Durban 4000, South Africa; nkambuleb@ukzn.ac.za (B.B.N.); dludlap@unizulu.ac.za (P.V.D.); 8Department of Biochemistry and Microbiology, University of Zululand, KwaDlangezwa 3880, South Africa

**Keywords:** cardiovascular disease, endothelial dysfunction, highly active antiretroviral therapy, human immunodeficiency virus, inflammation, myeloperoxidase, oxidative stress

## Abstract

People living with the human immunodeficiency virus (PLWH) are continually subjected to challenges involving the development of non-acquired immunodeficiency syndrome (AIDS)-related comorbidities despite the effectiveness of highly active antiretroviral therapy (HAART). Exacerbated oxidative stress, which is intrinsically linked to chronic inflammation, is implicated in non-AIDS comorbidities, including the increased risk of cardiovascular disease (CVD) observed in PLWH. Here, we review evidence on the potential pathological implications of myeloperoxidase (MPO), a leukocyte-derived enzyme and a key mediator of oxidative stress and inflammation, in driving CVD-related complications in PLWH. A systematic review approach was taken to identify relevant clinical studies through searches of Cochrane Libraries, PubMed, Web of Science, ScienceDirect, and Google Scholar, up to the 30 June 2025. The summarized data appraised clinical studies (n = 14) on adults (n = 1445) with a mean age of 45 years reporting on the association between MPO and enhanced lipid peroxidation marked by elevated concentrations of oxidized low-density lipoprotein cholesterol (oxLDL-C) in PLWH. Such results were consistent with elevated inflammatory markers, including high sensitivity C-reactive protein (hsCRP), which was also linked with endothelial dysfunction. There is a lack of evidence linking the duration of HAART to MPO levels or an increased risk of CVD. However, there is room to explore whether enhanced levels of oxLDL-C, in association with sustained MPO activation, could drive CVD risk in PLWH. The present review provides essential information on the pathological relevance of MPO in endothelial dysfunction and CVD risk in PLWH.

## 1. Introduction

The inception of highly active antiretroviral therapy (HAART) has shown significant efficacy in viral suppression and subsequent immune reconstitution in people living with the human immunodeficiency virus (PLWH) [1]. Although it remains effective, prolonged exposure to HAART has been associated with other negative outcomes, resulting from the pathological consequences of exacerbated oxidative stress and inflammation, which lead to endothelial dysfunction in PLWH [2,3]. Beyond the well-revised consequences of inflammation [4,5], this has propelled a growing interest in understanding the link between oxidative stress and the development of cardiovascular disease (CVD)-related complications in PLWH [6,7]. Notably, increased circulating levels of myeloperoxidase (MPO), a neutrophil-derived peroxidase enzyme, have already been linked with the pathogenesis of CVD even in PLWH [8,9,10]. Indeed, sustained activation of neutrophils in PLWH may lead to the excessive release of MPO, potentially causing damage to cellular structures by oxidizing circulating low-density lipoprotein cholesterol (LDL-C), which is predominantly seen in individuals with the metabolic syndrome [3,10,11]. This highlights a critical interplay between metabolic dysregulation and oxidative stress, that may contribute to the elevated CVD risk in PLWH.

Oxidative stress describes an imbalance between the production of reactive oxygen species (ROS) and the inability of the human body’s intracellular antioxidant response system to neutralize these reactive intermediates [8,9,10]. A key mechanism of endothelial dysfunction involves the excessive production of ROS, which promotes the “uncoupling” of endothelial nitric oxide synthase (eNOS). This pathological switch causes eNOS to produce superoxide instead of nitric oxide, exacerbating oxidative stress and compromising the integrity of blood vessels [12,13]. Existing research evidence suggests that MPO may contribute to oxidative stress-induced cellular damage within the endothelium, thereby increasing the risk of CVD in PLWH [14,15]. This underscores an urgent need to understand the precursors that induce oxidative stress and drive endothelial damage, seen in PLWH, especially those receiving HAART.

It is currently understood that MPO, through its oxidative activity, not only contributes to endothelial dysfunction but may also serve as a potential biomarker and therapeutic target in this high-risk population. Despite its emerging relevance [8,9,14,15], the precise contribution of MPO in driving CVD-related complications in PLWH remains incompletely understood. Therefore, the current study aims to systematically review existing literature on the involvement of MPO in promoting oxidative stress and its association with CVD risk among PLWH. Beyond employing a systematic review approach to ensure methodological rigor in literature selection and analysis, this study highlights the potential of MPO as a central player in inflammation-associated cardiovascular pathology. Importantly, a general overview of the peroxidase, MPO, is covered to highlight its potential role in driving oxidative stress-related CVD complications.

## 2. General Overview of Myeloperoxidase and Its Potential Pathological Role

### 2.1. The Discovery of Myeloperoxidase

Myeloperoxidase is an essential enzyme associated with innate immune responses and well recognized for its historical significance in cellular defense mechanisms [9,16]. Myeloperoxidase was first discovered in the mid-20th century, especially their involvement in the complexity of the immune system [17]. This heme-containing enzyme, primarily synthesized by neutrophils and monocytes, utilizes oxygen-dependent reactions to eradicate invading pathogens, including bacteria and viruses such as HIV [18,19]. Myeloperoxidase catalytic properties depend on its ability to utilize hydrogen peroxide (H_2_O_2_) and halides, such as chloride ions, to generate hypochlorous acid (HOCl), a potent antimicrobial agent crucial for pathogen eradication [9,16]. Interestingly, MPO’s oxidative role extends beyond the eradication of pathogens through its free radical scavenging activity [9,17], to include the oxidation of circulating lipids and proteins [9,20]. Myeloperoxidase’s activity is consistent with an abnormal inflammatory response that may result in tissue damage during disease progression [21,22,23]. Thus, sustained activation of MPO has been implicated in various inflammatory diseases, including endothelial dysfunction and atherosclerosis [24], highlighting its complex involvement in host defense and pathological processes.

### 2.2. The Pathological Link Between Myeloperoxidase and Endothelial Dysfunction

The involvement of MPO in the development of endothelial dysfunction is linked to its competency to oxidize polyunsaturated lipids, leading to the formation of reactive aldehydes, including malondialdehyde (MDA) [21,22,23]. MPO exhibits a dual role, for example, while it is essential for innate immunity through its antimicrobial activity, it can also act as a pro-oxidant [16,17,18]. Notably, MPO catalyzes the reaction of H_2_O_2_ with chloride ions to generate HOCl, a potent oxidizing agent [16,17,18]. HOCl subsequently reacts with polyunsaturated fatty acids in circulating lipids such as LDL-C, forming lipid peroxides and reactive aldehydes like 4-hydroxy-2-nonenal (4-HNE) and MDA, which contribute to oxidative stress and endothelial damage [16,17,18]. This pro-oxidant activity becomes particularly pathogenic under conditions of sustained neutrophil activation [16,17,18]. This process is consistent with the activation of neutrophils in response to invading pathogens, including HIV [2,3,23]. This activation triggers the release of MPO from intracellular granules into the extracellular environment or within phagosomes, where pathogens are engulfed. Alternatively, this process may accelerate the actions of nicotinamide adenine dinucleotide phosphate (NADPH) oxidase or other ROS-generating enzymes [25,26,27]. During this process, lipid intermediates react with molecular oxygen to form lipid peroxy radicals (LOO^•−^) [23,26], which damage cellular components and exacerbate oxidative stress [26]. Importantly, MPO’s antioxidant role is context-dependent, as its activity helps neutralize pathogens during acute immune responses, but chronic overactivation shifts its function towards tissue-damaging oxidation [16,17,18]. Thus, the unregulated activation of MPO can contribute to various pathological conditions, including oxidative stress, inflammation, and endothelial tissue damage [28], highlighting its delicate role in maintaining proper immune function.

### 2.3. Pathological Implications of Myeloperoxidase in PLWH

Relevant to PLWH who exhibit persistent immune activation and inflammation [3,29], elevated levels of MPO may drive the accumulation of oxidized lipids and ROS, which negatively impair endothelial nitric oxide bioavailability, contributing to vasoconstriction, atherosclerotic plaque formation, and possibly hypertension [30,31]. The MPO-mediated formation of oxLDL and reactive aldehydes, including MDA and 4-HNE, promotes oxidative modification of LDL particles, facilitating foam cell formation and early atherogenesis [16,17,18]. In addition, the heightened oxidant products in PLWH may exacerbate pre-existing metabolic risk factors, such as dyslipidemia and insulin resistance, leading to the development of CVDs [32,33]. Thus, understanding the precise pro-oxidant and context-dependent antioxidant roles of MPO provides insight into how its sustained activation contributes to CVD risk, while also highlighting its potential as a therapeutic target or biomarker in PLWH (Figure 1). 

## 3. Systematic Review Evidence Linking Myeloperoxidase Activity with Oxidative Stress and Inflammation-Associated Cardiovascular Complications in PLWH

### 3.1. An Overview of the Methodological Approach for a Systematic Review

This study took a systematic review approach following the Cochrane Preferred Reporting Items for Systematic Reviews and Meta-Analyses (PRISMA) guidelines [34]. A comprehensive literature search was performed across PubMed, Web of Science, ScienceDirect, Google Scholar, and the Cochrane Library up to June 2025, using a combination of keywords and medical subject headings (MeSH) including “myeloperoxidase (MPO)”, “peroxide”, “inflammation”, “endothelial dysfunction”, “HAART”, and “HIV”. Study quality and risk of bias were assessed by three reviewers using the National Institute of Health (NIH) quality of evidence risk assessment tool for observational studies and the Downs and Black tool for randomized trials. Interrater reliability was confirmed using the Intraclass Correlation Coefficient (ICC) in Statistical Package for the Social Sciences (SPSS) (version 29.0). Beyond giving the general characteristic features of included studies, the following sections cover critical information reporting on how MPO affects oxidative stress and inflammation through dysregulated lipid metabolism, which predisposes individuals to impaired cardiovascular function. We first looked at the role of MPO in oxidizing LDL-C, followed by its role in inflammation and endothelial dysfunction. Lastly, we examined the effects of HAART on MPO expression, at less than three years or more than three years of exposure to this antiretroviral therapy.

### 3.2. Characteristic Features of the Included Studies

The structured systematic search strategy of key terms and MeSH identified 14 clinical studies deemed eligible for inclusion in the present systematic review, reporting on the potential link between lipid peroxidation and CVD risk in PLWH. Collectively, the present study reports on 1445 participants (PLWH) with an average age of 45 years. The presented literature predominantly includes males (71.43%) of Caucasian ethnicity (62%) (Table 1). The majority of the included studies were from the United States of America (n = 7), while others were evenly distributed between Spain (n = 1), Denmark (n = 1), Zimbabwe (n = 1), Kenya (n = 1), Brazil (n = 1), China (n = 1), and Mexico (n = 1) (Table 1).

### 3.3. Abnormally Increased Low-Density Lipoprotein Levels Are Associated with Increased Myeloperoxidase Activity and Cardiovascular Disease Risk in PLWH

Dyslipidemia, through increased levels of LDL-C, is already an established consequence of endothelial dysfunction that may lead to increased CVD-risk [2,3]. Thus, it remained important to determine whether existing evidence links the dysregulation of MPO to the pathological consequences of oxidative stress and inflammation, which are predominant indicators of CVD-risk in PLWH [2,3]. Table 1 summarizes the evidence that indicates a strong association between increased levels of MPO and enhanced lipid peroxidation, coinciding with elevated concentrations of oxLDL-C in PLWH [40,44]. Moreover, this consequence was consistent with the literature, which indicates that elevated oxLDL-C may predict inflammation through increased levels of hsCRP and endothelial dysfunction, marked by reduced carotid intima-media thickness (cIMT) (Table 1) [35,37,40,44,45,46,47]. Interestingly, beyond oxLDL-C, the dysregulation of other lipid profiles such as high-density lipoprotein-cholesterol (HDL-C), triglycerides, and total cholesterol may also predict increased CVD risk in PLWH [38,42,44]. Such evidence indicates that an impaired metabolic state, likely characterized by abnormal lipid levels like oxLDL-C, is associated with endothelial dysfunction and may predict CVD risk in PLWH. However, confirmatory studies of these effects are required in a larger cohort, also reporting on established markers of endothelial dysfunction like soluble intercellular and vascular adhesion molecule-1 (sICAM-1 and sVCAM-1), soluble endothelial selectin (sE-selectin), flow-mediated dilation (FMD), and cIMT (Table 1).

### 3.4. Pro-Inflammatory Markers Associate Myeloperoxidase Activity with Endothelial Dysfunction in PLWH

Inflammation, including its pathological role, is a well-studied phenomenon in PLWH [44,49,50]. Beyond the implications of hsCRP during sustained activation of MPO [46], the expression and release of pro-inflammatory cytokines, including IL-15 and interferon gamma, were associated with the exacerbation of oxidative stress-related cellular damage within a pathological state [45]. Table 1 shows six studies outlining a link between enhanced markers of inflammation, which may predict endothelial dysfunction in PLWH. For example, the two studies reported an abnormal elevation of pro-inflammatory cytokines, including cluster of differentiation (CD)-14, CD141, and IL-15, in PLWH on HAART, along with MPO [45,47]. The pro-inflammatory state in PLWH provides a clear link between MPO and endothelial dysfunction. Elevated levels of chemoattractants like IL-8 and monocyte chemotactic protein-1 signal the persistent trafficking of MPO-rich leukocytes to the vasculature [36,42]. The subsequent release of MPO can then induce a state of endothelial activation, characterized by the increased expression of adhesion molecules such as VCAM-1 and ICAM-1 [51,52]. These adhesion molecules are not only established markers of endothelial dysfunction but also facilitate further recruitment of inflammatory cells [2,3]. Thus, a feedback loop is created where circulating pro-inflammatory markers promote MPO release, and its activity, in turn, amplifies the inflammatory state of the endothelium itself. However, beyond the observed abnormalities in adhesive molecules that indicate a pro-inflammatory state, there is little evidence available linking MPO and inflammation to endothelial dysfunction in PLWH.

### 3.5. Evidence Showing the Potential Influence of HAART on Myeloperoxidase Activity in PLWH

Understandably, HAART has been previously associated with alterations in various aspects of immune function, including changes in lipid peroxidation markers such as MPO [41]. Notably, the contributory effects of HAART on sustained activation of MPO may vary depending on factors such as regimen combination, duration of therapy, and individual patient characteristics. This may be further influenced by specific antiretroviral drugs such as atazanavir-boosted-ritonavir (ATV-r), zidovudine (AZT), and lamivudine (3TC) [41,46,48]. Table 1 indicates that seven studies support the notion that PLWH on HAART display an increased expression of inflammatory markers, which may be associated with elevated levels of MPO [35,37,42,45,46,47]. The addition of protease inhibitors, which inhibit the replication of HIV, were shown to elevate oxLDL-C (potentially induced by elevating MPO), despite being administered together with a lipid-lowering drug, rosuvastatin [37]. This was later supported by Wang et al. (2020) [45] who observed elevated levels of MPO in PLWH on HAART; however, these authors did not disclose the HAART regimen administered to the study participants. Similar observations of elevated MPO were made in PLWH actively administering the NRTI-based HAART regimen [43,46,48]. Despite such evidence, the majority (n = 10) of the included studies failed to disclose the direct effects of HAART on modulating both lipid peroxidation and endothelial markers. This suggests that further research is necessary to elucidate how various HAART regimens impact MPO activity or related oxidative stress markers in PLWH.

### 3.6. Myeloperoxidase Activity and Endothelial Dysfunction in Individuals Exposed to HAART for Less than Three-Years

The duration of HAART exposure in PLWH was another critical aspect investigated in the present study. Among the included literature, five studies reported on PLWH exposed to HAART for less than three-years (Table 1). Only one study failed to disclose the duration of participants’ treatment [45]. Three studies reported no significant association between the use of NRTIs, non-nucleoside reverse transcriptase inhibitor (NNRTIs), protease (PIs), or integrase strand transfer inhibitors (INSTIs) for less than three years and elevated MPO or lipid peroxidation, and endothelial dysfunction in PLWH [35,47,48]. Highly Active Antiretroviral Therapy exposure for at least two months has been shown to indicate that NNRTIs may reduce peroxide in PLWH [35]. Highly active antiretroviral therapy exposure duration of at least three-months indicated that NRTI-based regimens were linked to elevated MPO levels and reduced mitochondrial phosphorylation activity, known contributors to the production of ROS, major risk factors for CVDs [43]. These observations were partly supported by Mitchell et al., (2020) [44] who noted an association between oxLDL-C, highlighting that oxLDL formation can be induced by circulating levels of MPO in PLWH undergoing HAART. Following a six-month exposure duration, HAART containing either NNRTIs, PIs, or INSTIs may potentially exacerbate endothelial dysfunction through the pathological consequences of inflammation and enhanced formation of oxLDL-C and hsCRP expression, impaired cIMT, and subsequently dysregulate lipid metabolism leading to the manifestation of CVDs in PLWH [37,40,42]. The data extracted from the included studies present contradicting findings when looking at a less than 3-year duration of exposure to HAART in modulating MPO-related endothelial dysfunction, warranting further research to confirm these findings.

### 3.7. Myeloperoxidase Activity and Endothelial Dysfunction Following Three 3-Years or More Exposure Duration to HAART

The potential role of prolonged HAART duration was set at three years or more, looking at potential detrimental consequences linking MPO activation with CVD-related risk. Notably, seven studies reported on participants with a HAART exposure duration of at least three-years or more (Table 1). The initial report indicated that NRTI, NNRTI, INSTI or protease inhibitor-based HAART exposure for at least three years was associated with raised levels of MPO that are consistent with increased markers of endothelial dysfunction, such as sE-selectin, sICAM-1, and sVCAM-1 in PLWH [36,39]. This finding was consistent with a report indicating that a four-year NNRTI or PI-based HAART exposure duration was associated with reduced levels of adiponectin, a cardioprotective adipokine, as well as elevated concentrations of LDL-C subclasses in PLWH [41]. However, these studies observed that an increase in both MPO and markers of endothelial dysfunction was not linked to adverse cardiovascular outcomes, including myocardial infarction and coronary artery disease [36,38,39,41]. Alternatively, others [48] observed that 4-years of exposure to NNRTI or PI-based HAART appears to be associated with reduced circulating levels of MPO and endothelial dysfunction markers such as sICAM-1 and sVCAM-1 in PLWH, comparable to those observed in HIV-negative persons. Notably, only one study reported on a prolonged exposure to HAART that exceeded 5 years with a duration of 10-years and did not observe any association between MPO and endothelial dysfunction markers, including sE-selectin, sICAM-1, and sVCAM-1, with adverse cardiovascular outcomes, including myocardial infarction in PLWH [36]. These conflicting findings make it essential to understand the interplay between HAART and its exposure duration with oxidative stress and cardiovascular risk in PLWH.

### 3.8. Quality of Evidence and Risk of Bias of the Included Studies

The current study utilized the National Institutes of Health quality assessment tool to evaluate the quality of evidence and risk of bias for the included studies [53]. The results indicate that most of the included studies (n = 13) reported good quality of evidence and risk of bias after scoring 9–11 points out of a possible score of 14. Only one study reported an excellent quality of evidence and low risk of bias, scoring 12 points out of a possible score of 14. The interrater agreement between the three independent authors was determined using the Intraclass Correlation Coefficient (ICC) statistical test. The results indicate that rater-1 gave the highest rating score for the included studies, with a mean (±SD) of 10.79 ± 0.699. This was followed by rater-3, with a mean (±SD) of 10.36 ± 0.929. Rater 2 gave the lowest rating score with a mean (±SD) of 10.29 ± 1.267. The overall ICC value for the three raters was 0.553 at *p* = 0.036, indicating a moderate agreement and a reliable quality of evidence and risk of bias reported by the included studies.

## 4. Discussion

This review set out to address the limited understanding of how oxidative stress markers, particularly MPO, contribute to cardiovascular risk among PLWH on HAART. This review demonstrates the need to better understand a potential pathological link between oxidative stress, lipid profiles, inflammation, endothelial dysfunction, and CVD risk. These insights may help uncover relevant clinical biomarkers to monitor this high-risk population. The findings of the present review highlight the complex interplay between oxidative stress markers, lipid profiles, inflammation, endothelial dysfunction, and CVD-risk in PLWH on HAART. Elevated plasma peroxide levels were positively associated with inflammatory (hsCRP) and lipid (LDL-C) markers but inversely associated with traditional CVD risk factors such as age and BMI [35,37,44,46], suggesting that oxidative stress may act as an early or independent marker of cardiovascular pathology. Notably, individuals on NNRTI-based regimens exhibited significantly lower peroxide levels compared to those receiving regimens comprising PIs, indicating a potentially more favorable cardiovascular risk profile.

Although neutrophils are primarily antibacterial, HIV infection induces chronic immune activation, stimulating neutrophils via cytokines such as IL-6, IL-8, IL-1β, and TNF-α released from pyroptotic CD4+ T-cells [10,14,19,20]. This leads to MPO release and oxidative stress even without direct viral targeting. Additionally, HIV-associated neutropenia may reduce absolute neutrophil counts, but the remaining neutrophils often exhibit hyperactivation, contributing to endothelial dysfunction and lipid oxidation [10,14,19,20].

Although MPO and markers of endothelial dysfunction (sICAM-1, sVCAM-1, sE-selectin) were often elevated, they were not consistently associated with clinical endpoints such as myocardial infarction or impaired FMD. However, MPO contributes to endothelial dysfunction primarily through its pro-oxidant activity, generating reactive species such as HOCl that oxidize LDL-C to form oxLDL-C [16,17,18]. Notably, oxLDL-C triggers vascular inflammation, recruitment of monocytes/macrophages, and foam cell formation, collectively impairing endothelial integrity and function [16,17,18]. Therefore, sustained MPO activation not only reflects oxidative stress but directly mediates vascular injury, linking chronic inflammation with early atherogenesis in PLWH on HAART [37,41,44]. Similarly, oxHDL-C was associated with higher BMI and lower apolipoprotein (Apo-AI), indicating altered lipid metabolism and potential CVD risk [40,42]. These findings suggest that oxidative stress and lipoprotein modifications may be early contributors to endothelial dysfunction, independent of traditional lipid metrics.

The review also reveals variable effects of HAART regimens on oxidative stress, particularly the role of NRTIs such as abacavir, which are associated with increased MPO levels and oxidative stress within months of HAART initiation. While MPO and inflammatory markers such as CD14, IL-15, and interferon-gamma were elevated in some studies involving NNRTIs and PIs, the direct modulatory effects of specific antiretroviral drugs remain unclear [45,47]. Notably, different drug classes appear to have divergent effects on MPO activity, for example, NRTIs like abacavir may increase MPO-mediated oxidative stress, PIs are also associated with elevated oxidative markers, whereas NNRTIs and some INSTIs may exert comparatively lower MPO activation. This suggests that the choice of HAART regimen could influence cardiovascular risk via differential impacts on oxidative stress and endothelial function [35,39,47,48]. Interestingly, prolonged HAART exposure was associated with reduced endothelial dysfunction markers in some studies, suggesting a potential time-dependent regulatory effect [35,39,47,48]. Despite MPO being independently linked to reduced cIMT, other markers, such as LDL-C subclass B and adiponectin, showed a stronger predictive value for vascular health [41,42].

Overall, these findings underscore the importance of considering oxidative, inflammatory, and endothelial dysfunction markers in CVD risk stratification among PLWH. However, their predictive utility may be enhanced when evaluated alongside lipid subfractions and adipokines. These findings also hold potential clinical implications. For example, monitoring specific biomarkers such as MPO or oxLDL-C could complement traditional risk assessments by helping to identify PLWH who may be at heightened cardiovascular risk despite relatively normal lipid profiles. Such early detection may enable more targeted interventions or closer clinical monitoring, although further validation in large prospective studies is warranted. Nonetheless, traditional and lifestyle-related risk factors, including age, ethnicity, smoking, alcohol use, diet, genetic predisposition, and co-infections, must not be overlooked in the multifactorial development of CVD in this population [3,54,55].

## 5. Study Limitations and Strengths

The use of secondary data remains a significant limitation of this study, in addition to retrieving fewer studies meeting our inclusion criteria, with varying study designs. Considerable heterogeneity across the included studies, particularly in patient populations, HAART regimens, and the assays used to measure oxidative stress and endothelial dysfunction, likely contributed to the inconsistent associations observed. Although the outcomes of the included studies were comparable and could be pooled purposefully, the interventions and comparators across the incorporated studies differed, which limited us in fully addressing the impact of specific HAART drug combinations on MPO-driven lipid peroxidation. Despite these limitations, this review adheres to the PRISMA guidelines to comprehensively retrieve studies reporting on MPO-mediated lipid peroxidation and the risk of developing CVDs in PLWH. This allowed us to systematically extract and compare evidence from existing literature purposefully. The qualitative approach in analyzing the current data brings strength by taking into consideration the many factors presented by each study before conclusions are drawn. Future research should therefore aim to reduce variability by employing standardized assays and recruiting more homogenous patient groupings to improve comparability and reproducibility. The evidence presented in this review gives a foundation for future research addressing the pathogenesis of CVDs in PLWH, especially when looking at the role of oxidative stress in the development of CVDs.

## 6. Conclusions

Despite the success of HAART in suppressing the viral load in PLWH, there remains scope to reduce the potential contribution of HAART regimens in the pathogenesis of CVDs. This could, in part, be explained by the impaired lipid and glucose metabolic states, in conjunction with disrupted redox pathways. Existing evidence shows that increased levels of MPO coincide with sustained inflammation and elevated oxLDL-C, which may likely contribute to endothelial dysfunction and subsequent CVDs. In the present study, we acknowledge the existence of conflicting findings regarding the effects of MPO-mediated lipid peroxidation and the development of CVDs in PLWH, who are actively receiving HAART. The current results are based on limited evidence; therefore, more research is required to understand the pathophysiology of lipid peroxidation and endothelial dysfunction preceding the development of CVDs in PLWH receiving HAART. Importantly, identifying reliable biomarkers such as MPO, oxLDL-C, or related oxidative and inflammatory markers could support early detection and risk stratification in high-risk patients. Future investigations should adopt more targeted approaches, particularly prospective cohort studies examining specific antiretroviral drug regimens, and expand beyond MPO by leveraging multi-omics biomarker networks that may identify novel targets, enhance risk prediction and early intervention strategies.

While antiretroviral therapy effectively controls HIV, certain regimens may contribute to cardiovascular risk through sustained oxidative stress and inflammation. Clinicians should remain vigilant, and further research is needed to optimize treatment strategies and minimize potential cardiovascular complications.

## Figures and Tables

**Figure 1 ijms-26-10285-f001:**
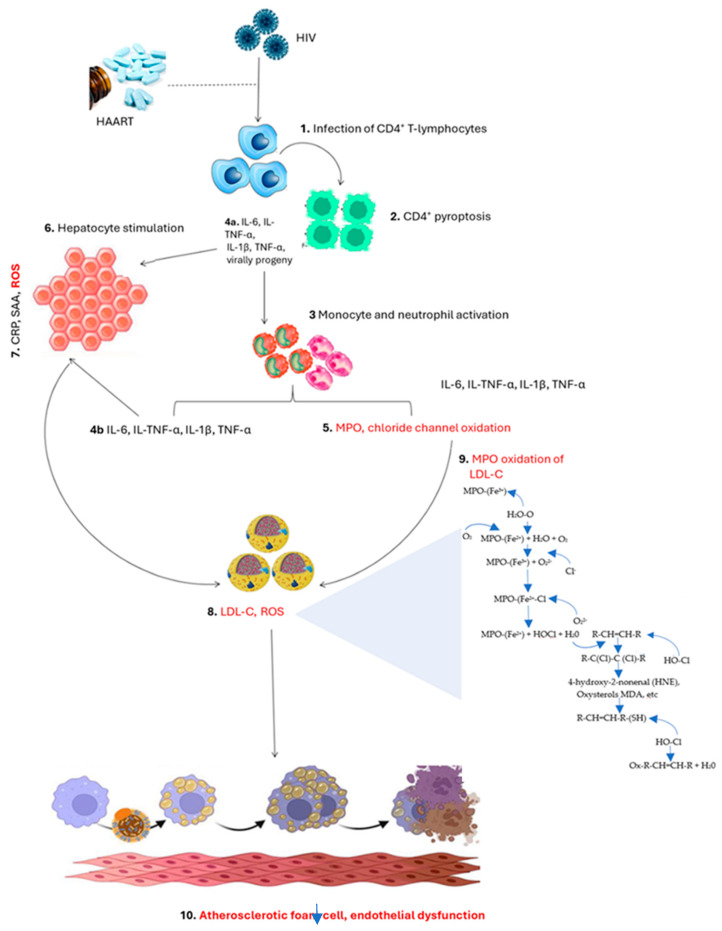
An overview pathway showing myeloperoxidase (MPO) and its associated role during the pathogenesis linking lipid peroxidation and endothelial dysfunction. Briefly, the human immunodeficiency virus infects cluster differentiation 4 positive (CD4+) T-lymphocytes, causing them to undergo pyroptosis. During pyroptosis, dying CD4+ T-cells and tissue-resident macrophages release pro-inflammatory cytokines including interleukin (IL)-6, IL-8, IL-1β, and tumor necrosis factor-alpha (TNF-α). These cytokines act in a paracrine manner to activate circulating monocytes and neutrophils. In neutrophils, cytokine signaling triggers degranulation and MPO release, while in monocytes, it promotes differentiation into macrophages and enhances oxidative burst activity. In addition, these inflammatory cytokines activate chloride channels in endothelial cells and other tissues. Activated monocytes and neutrophils release MPO and more cytokines, which sensitize hepatocytes to generate and release acute-phase proteins such as C-reactive protein (CRP) and serum amyloid A (SAA) in conjunction with reactive oxygen species (ROS) and low-density lipoprotein cholesterol (LDL-C). Circulating chloride ions reacts with ROS such as hydrogen peroxide (H_2_O_2_), catalyzed by MPO, to form hypochlorous acid (HOCl) through several steps, which generate aldehyde intermediates including 4-hydroxy-2-nonenal (HNE), oxysterols, and malondialdehyde (MDA). This illustrates the pro-oxidant role of MPO, contributing to oxidative stress, lipid peroxidation, and endothelial dysfunction. HOCl-mediated oxidation of LDL-C produces oxidized LDL-C (oxLDL-C), promoting monocyte and macrophage infiltration into the endothelium, foam cell formation, and subsequent endothelial injury. MPO also has context-dependent antioxidant functions, neutralizing pathogens during acute immune responses, but chronic overactivation shifts its activity towards tissue-damaging oxidation. In response, the immune system expresses monocytes and macrophages to eliminate oxLDL-C (oxLDL-C). Unfortunately, these immune cells are converted into atherosclerotic foam cells during this process, which contributes to endothelial dysfunction (Created using Microsoft PowerPoint, 2021).

**Table 1 ijms-26-10285-t001:** Evidence on the potential link between myeloperoxidase and cardiovascular disease -risk factors in people living with the human immunodeficiency virus.

Authors, Year	Region of Study	Study Design	Population	Intervention	Key Observations
Masiá et al., 2007 [35]	Spain	Cross-sectional	People living with the human immunodeficiency virus (PLWH) (n = 181) with an average age of 40-years, predominately males (79%) of Caucasian (97%) ethnicity.	Non-nucleoside reverse transcriptase inhibitor (NNRTI), protease inhibitor (PI), nucleotide reverse transcriptase inhibitor (NRTI), or NNRTI plus PI. Participants were on highly active antiretroviral therapy (HAART) for at least two months.	Plasma peroxide levels were positively associated with high-sensitivity C-reactive protein (hsCRP) and low-density lipoprotein-cholesterol (LDL-C) but negatively associated with traditional cardiovascular disease (CVD)-risk factors such as age and body mass index (BMI). NNRTI-based regimen significantly lowered peroxide levels as indicators of CVD-risk when compared to PIs.
Knudsen et al., 2014 [36]	Denmark	Case–control	PLWH (n = 108) with an average age of 49-years, strictly males (100%) of Caucasian (100%) ethnicity.	NNRTI, PI, NRTI, or NNRTI plus PI. Participants were on HAART for ten years.	Plasma peroxide inducer myeloperoxidase (MPO) and markers of endothelial dysfunction, including sE-selectin, soluble intercellular- and vascular adhesion molecules (sICAM-1 and sVCAM-1), were not associated with myocardial infarction.
Hileman et al., 2016 [37]	USA	Randomized controlled trial	PLWH (n = 147) with an average age of 45-years, predominately males (78%) of African (68%) ethnicity.	PIs plus 10 mg of rosuvastatin daily. Participants on HAART were monitored for six months.	Plasma-oxidized LDL-C (oxLDL-C) was positively linked with hsCRP and changes in endothelial function as measured by carotid intima-media thickness (cIMT).
Zhou et al., 2016 [38]	Zimbabwe	Observational	PLWH (n = 147) with an average age of 41-years, predominately females (76.7%) of African (100%) ethnicity.	NNRTI or PIs. Participants were on HAART for at least one month but were allowed to change treatment during the nine months of the study period.	Serum elevated MPO levels were linked with LDL-C but not coronary artery disease.
Zungsontiporn et al., 2016 [39]	USA	Cross-sectional	PLWH (n = 135) with an average age of 50- years, predominately males (88.1%) of Caucasian (58.5%) ethnicity.	NRTI, NNRTI, PIs, or integrase strand transfer inhibitors (INSTIs). Participants were on HAART for three years or more.	Plasma MPO levels and markers of endothelial dysfunction including sE-selectin, sICAM-1, and sVCAM-1 were not linked with brachial artery flow mediated dilation (FMD).
Kelesidis et al., 2017 [40]	USA	Cross-sectional	PLWH (n = 116) with an average age of 48- years, predominately males (100%) of Caucasian (67%) ethnicity.	NNRTIs, PIs, or INSTIs. Participants were on HAART for at least six months.	Plasma-oxidized high-density lipoprotein cholesterol (oxHDL-C) was associated with a higher BMI, lower apolipoprotein-AI, and CVD-risk.
Lui et al., 2017 [41]	China	Observational	PLWH (n = 61) with an average age of 50-years, predominately males (89%) of Chinese (97%) ethnicity.	NNRTIs or PIs. Participants were on HAART for four years.	Plasma MPO was independently associated with endothelial dysfunction marked by reduced cIMT. However, LDL-C subclass pattern type B and adiponectin levels were the best predictors of reduced cIMT progression.
Weke et al., 2018 [42]	Kenya	Cross-sectional	PLWH (n = 120) with an average age of 34-years, predominately females (64.2%) of African (100%) ethnicity.	HAART-regimen not disclosed. Participants were on HAART for at least six months to five years.	Serum MPO and lipoprotein-associated phospholipase A2 (Lp-PLA2) were elevated and predicted early events of developing CVDs.
Gangcuangco et al., 2020 [43]	USA	Cross-sectional	PLWH (n = 149) with an average age of 51-years, predominately males (88.6%) of Caucasian (57.7%) ethnicity.	NRTIs. Participants were on HAART for three months or more.	Plasma MPO levels were elevated and linked with low mitochondrial oxidative phosphorylation, and current use of NRTIs. Moreover, low mitochondrial phosphorylation was linked with sICAM-1, a marker of endothelial dysfunction.
Mitchell et al., 2020 [44]	USA	Observational	PLWH (n = 33) with an average age of 53-years, predominately males (88%) of Caucasian (67%) ethnicity.	NRTI, NNRTI, or PIs. Participants were on HAART for three months or more.	Plasma oxLDL-C was associated with impaired LDL-C, HDL-C, triglycerides, and total cholesterol, known CVD-risk factors.
Wang et al., 2020 [45]	USA	Observational	PLWH (=50). The demographic data of the study population was not disclosed.	PIs. Participant’s HAART duration not disclosed.	Plasma MPO levels were elevated and consistent with an increase in inflammation, marked by plasma CD14, Interleukin-15, and interferon gamma.
Borato et al., 2022 [46]	Brazil	Cross-sectional	PLWH (n = 104) with an average age of 41-years, predominately females (62%) of Caucasian (100%) ethnicity.	NRTIs, NNRTIs, or PIs. Participants were on HAART for six years.	Serum MPO and hsCRP as predictors of CVD-risk were elevated. MPO was elevated in the presence of NNRTIs and PIs, with NRTIs (abacavir) intensifying these levels following three months of administration.
De Menezes et al., 2022 [47]	USA	Observational	PLWH (n = 74) having existing CVDs with an average age of 50-years, predominately males (92%) of Caucasian (100%) ethnicity.	HAART not disclosed. Participants were monitored for five years.	Plasma MPO was associated with CD141, and CD14 extracellular vesicles, a predictor of CVDs. Moreover, total extracellular vesicle count was elevated. The direct effects of HAART in modulating these markers were not disclosed.
Martínez-Ayala et al., 2023 [48]	Mexico	Cross-sectional	PLWH (n = 20) with an average age of 35-years, predominately males (95%) of Hispanic (100%) ethnicity.	NNRTIs or PIs. Participants were on HAART for four years.	Serum MPO and markers of endothelial dysfunction including sICAM-1, and sVCAM-1 were not significantly different between groups. HAART had a greater impact in reducing MPO, sICAM-1, and sVCAM-1.

## Data Availability

No new data were created or analyzed in this study. Data sharing is not applicable to this article.

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
