# Peer review of "Sustained Activation of Myeloperoxidase Is Associated with Oxidative Stress and Inflammation in People Living with the Human Immunodeficiency Virus at Risk of Cardiovascular Disease"

_ijms, 2025, doi:10.3390/ijms262110285_

Round 1

Reviewer 1 Report

Comments and Suggestions for Authors

This manuscript explores the potential pathological mechanisms by which MPO may drive oxidative stress and chronic inflammation, consequently contributing to the increased risk of CVD in people living with PLWH. It discusses the differential impacts of various HAART regimens (e.g., NNRTIs vs. PIs) on oxidative stress levels. The review synthesizes existing evidence, clearly identifies current gaps and contradictions in the field, and provides a foundational basis for future research directions. Specific comments and suggestions for revision are as follows:

1.The title "Sustained activation of myeloperoxidase may drive..." strongly emphasizes the "driving" role of MPO. However, the discussion and conclusion sections note that the association between MPO and clinical endpoints (e.g., myocardial infarction) is inconsistent, and its predictive value may be lower than other indicators (e.g., LDL-C subfractions, adiponectin). It is suggested whether the expression of the title could be modified to better reflect these nuanced findings.

2.The opening of the Discussion section could be fine-tuned to not merely reiterate the findings, but to directly highlight how these findings address the research questions or gaps identified in the Introduction.

3.The potential implications of these findings for clinical practice could be briefly discussed; for instance, whether monitoring MPO or oxLDL-C could aid in early intervention for specific high-risk patients.

4.The elaboration on study limitations could be strengthened by emphasizing that the heterogeneity of the included studies is a likely reason for many inconsistent associations and by calling for future research to employ standardized assays and more homogenous patient groupings.

5.Future research directions should be more targeted, advocating for prospective cohort studies focused on specific drug regimens and exploring the role of multi-omics biomarker networks beyond MPO in risk prediction.

6.The strength of the conclusions should be carefully reviewed to ensure the tone matches the "uncertainty" and "limited evidence" presented in the manuscript, avoiding overinterpretation that extends beyond the support provided by the data.

Author Response

Reviewer 1

This manuscript explores the potential pathological mechanisms by which MPO may drive oxidative stress and chronic inflammation, consequently contributing to the increased risk of CVD in people living with PLWH. It discusses the differential impacts of various HAART regimens (e.g., NNRTIs vs. PIs) on oxidative stress levels. The review synthesizes existing evidence, clearly identifies current gaps and contradictions in the field, and provides a foundational basis for future research directions. Specific comments and suggestions for revision are as follows:

We sincerely thank the reviewer for their valuable feedback and constructive suggestions, which have significantly improved the quality of our manuscript. We have carefully addressed all the comments and incorporated the recommended revisions, including enhancing the clarity and flow of information, improving readability, and modifying major aspects such as the title to better reflect the associations studied. All changes have been clearly highlighted using track changes for ease of review. We hope that the revised manuscript meets your expectations and is now suitable for publication.

1.The title "Sustained activation of myeloperoxidase may drive..." strongly emphasizes the "driving" role of MPO. However, the discussion and conclusion sections note that the association between MPO and clinical endpoints (e.g., myocardial infarction) is inconsistent, and its predictive value may be lower than other indicators (e.g., LDL-C subfractions, adiponectin). It is suggested whether the expression of the title could be modified to better reflect these nuanced findings.

We agree with the reviewer, as such we the title “Sustained activation of myeloperoxidase may drive oxidative stress and inflammation-associated cardiovascular complications in people living with the human immunodeficiency virus.” Has been changed to “Sustained activation of myeloperoxidase is associated with oxidative stress and inflammation in people living with the human immunodeficiency virus at risk of cardiovascular disease”

2.The opening of the Discussion section could be fine-tuned to not merely reiterate the findings, but to directly highlight how these findings address the research questions or gaps identified in the Introduction.

We agree with the reviewer, and the first paragraph of the discussion has been modified to clearly indicate pressing research questions and gaps in evidence as highlighted in the introduction. The newly added information, to introduce the discussion section “This review set out to address the limited understanding of how oxidative stress markers, particularly MPO, contribute to cardiovascular risk among PLWH on HAART. This review demonstrates the need to better understand a potential pathological link between oxidative stress, lipid profiles, inflammation, endothelial dysfunction, and CVD risk. These insights may help uncover clinical biomarkers in this high-risk population.”

3.The potential implications of these findings for clinical practice could be briefly discussed; for instance, whether monitoring MPO or oxLDL-C could aid in early intervention for specific high-risk patients.

We agree with the reviewer and appreciate this important comment. We have indeed modified the last paragraph of the discussion to highlight the reviewer’s comment. Newly added information “These findings also hold potential clinical implications. For example, monitoring specific biomarkers such as MPO or oxLDL-C could complement traditional risk assessments by helping to identify PLWH who may be at heightened cardiovascular risk despite relatively normal lipid profiles. Such early detection may enable more targeted interventions or closer clinical monitoring, although further validation in large prospective studies is warranted.”

4.The elaboration on study limitations could be strengthened by emphasizing that the heterogeneity of the included studies is a likely reason for many inconsistent associations and by calling for future research to employ standardized assays and more homogenous patient groupings.

Thank you for this important comment, that we have revised accordingly. We have revised the “Study limitations and strengths” section accordingly, inserting important information such as “Considerable heterogeneity across the included studies, particularly in patient populations, HAART regimens, and the assays used to measure oxidative stress and endothelial dysfunction, likely contributed to the inconsistent associations observed” and “Future research should therefore aim to reduce variability by employing standardized assays and recruiting more homogenous patient groupings to improve comparability and reproducibility.” to be consistent with what is recommended by the reviewer.

5.Future research directions should be more targeted, advocating for prospective cohort studies focused on specific drug regimens and exploring the role of multi-omics biomarker networks beyond MPO in risk prediction.

We thank the reviewer for this important comment, and we have modified the conclusion to indicate “Future investigations should adopt more targeted approaches, particularly prospective cohort studies examining specific antiretroviral drug regimens, and expand beyond MPO by leveraging multi-omics biomarker networks that may identify novel targets, enhance risk prediction and early intervention strategies.”

6.The strength of the conclusions should be carefully reviewed to ensure the tone matches the "uncertainty" and "limited evidence" presented in the manuscript, avoiding overinterpretation that extends beyond the support provided by the data.

We have revised the conclusion to explicitly address the reviewer’s suggestion by adding a statement on the potential clinical implications of the findings and more targeted future research directions. Specifically, we highlighted that identifying reliable biomarkers, such as MPO and oxLDL-C, could support early detection and risk stratification in high-risk PLWH. Additionally, we emphasized the importance of prospective cohort studies focusing on specific antiretroviral drug regimens, standardizing biomarker assays, and exploring multi-omics networks beyond MPO to improve risk prediction and guide personalized interventions.

Reviewer 2 Report

Comments and Suggestions for Authors

This is a systematic review that addresses a significant clinical issue: the increased risk of cardiovascular disease (CVD) in people living with HIV (PLWH) despite effective antiretroviral therapy. The manuscript is generally well-written, and PRISMA guidelines were utilized to analyze literature data, with focus on myeloperoxidase (MPO) as a potential mechanistic driver linking inflammation, oxidative stress and CVD in people living with HIV.

Since this is a review, the material should be presented in a way that is understandable to a wide audience. Some points are described by the authors very superficially and vaguely, they need to be improved

Comments

  1. Please, describe the pro-oxidant and antioxidant roles of myeloperoxidase more clearly. In particular, the MPO-mediated mechanisms of oxidation of polyunsaturated lipids should be described in detail. In the present form, the description of these processes is something contradictory.
  2. Neutrophils are a line of defense against bacteria, but not viruses. Activation of neutrophils under HIV infection should be explained in detail. In addition, a decrease in neutrophil levels, a condition known as neutropenia, is more commonly observed in HIV infection.
  3. Fig 1. “Briefly, the human immunodeficiency virus infects cluster differentiation 4 positive (CD4+) T-lymphocytes, causing them to undergo pyroptosis. This subsequently releases viral progeny and inflammatory cytokines including interleukin (IL)-6, IL-8, (IL-1β), and tumour necrosis factor-alpha (TNF-α), which activate monocytes and neutrophils”. Which cells produce cytokines? Please explain the mechanisms of activation of neutrophils and monocytes.
  4. The connection of MPO with endothelial dysfunction is not clearly presented.
  5. The specific effects of different drug classes (e.g., NRTIs like abacavir, Protease Inhibitors, INSTIs) on MPO activity should be discussed, as the results suggest they may have divergent effects

Author Response

Reviewer 2

This is a systematic review that addresses a significant clinical issue: the increased risk of cardiovascular disease (CVD) in people living with HIV (PLWH) despite effective antiretroviral therapy. The manuscript is generally well-written, and PRISMA guidelines were utilized to analyze literature data, with focus on myeloperoxidase (MPO) as a potential mechanistic driver linking inflammation, oxidative stress and CVD in people living with HIV. Since this is a review, the material should be presented in a way that is understandable to a wide audience. Some points are described by the authors very superficially and vaguely, they need to be improved.

We sincerely thank the reviewer for their valuable feedback and constructive suggestions, which have significantly improved the quality of our manuscript. We have addressed all comments by clarifying the pro-oxidant and context-dependent antioxidant roles of MPO, detailing neutrophil and monocyte activation in HIV, linking MPO activity to endothelial dysfunction, specifying cytokine sources in Figure 1, and discussing the differential effects of HAART drug classes on MPO and oxidative stress. All changes have been clearly highlighted using track changes for ease of review. We hope that the revised manuscript meets your expectations and is now suitable for publication.

Comments

  1. Please, describe the pro-oxidant and antioxidant roles of myeloperoxidase more clearly. In particular, the MPO-mediated mechanisms of oxidation of polyunsaturated lipids should be described in detail. In the present form, the description of these processes is something contradictory.

We thank the reviewer for this important observation. We have revised the manuscript and figure legend to clarify both the pro-oxidant and context-dependent antioxidant roles of MPO. Specifically, we now describe in detail how MPO catalyzes the reaction of hydrogen peroxide with chloride ions to form hypochlorous acid (HOCl), leading to oxidation of polyunsaturated lipids, formation of aldehyde intermediates (e.g., 4-hydroxy-2-nonenal, malondialdehyde), and subsequent generation of oxidized LDL-C (oxLDL-C). At the same time, we note that MPO possesses context-dependent antioxidant activity, helping to neutralize pathogens during acute immune responses, while chronic overactivation shifts its activity toward tissue-damaging oxidative processes. These revisions provide a more mechanistic understanding of MPO’s dual roles in oxidative stress and lipid peroxidation.

  1. Neutrophils are a line of defense against bacteria, but not viruses. Activation of neutrophils under HIV infection should be explained in detail. In addition, a decrease in neutrophil levels, a condition known as neutropenia, is more commonly observed in HIV infection.

We thank the reviewer for highlighting the need to clarify neutrophil activation under HIV infection. We have added a description of how chronic immune activation in PLWH stimulates neutrophils through cytokines released from pyroptotic CD4+ T-cells, leading to MPO release and oxidative stress. We also acknowledge the role of HIV-associated neutropenia, noting that while absolute neutrophil counts may decrease, the remaining neutrophils can exhibit hyperactivation that contributes to endothelial dysfunction and lipid oxidation. These additions are highlighted in the discussion section within the revised manuscript.

  1. Fig 1. “Briefly, the human immunodeficiency virus infects cluster differentiation 4 positive (CD4+) T-lymphocytes, causing them to undergo pyroptosis. This subsequently releases viral progeny and inflammatory cytokines including interleukin (IL)-6, IL-8, (IL-1β), and tumour necrosis factor-alpha (TNF-α), which activate monocytes and neutrophils”. Which cells produce cytokines? Please explain the mechanisms of activation of neutrophils and monocytes.

We thank the reviewer for requesting clarification regarding cytokine sources and the mechanisms of neutrophil and monocyte activation. In the revised figure legend, we specify that both pyroptotic CD4+ T-cells and tissue-resident macrophages release cytokines, which activate circulating neutrophils and monocytes through paracrine signaling. We also clarified that cytokine signaling in neutrophils triggers degranulation and MPO release, while in monocytes it promotes differentiation into macrophages and enhances oxidative burst activity. These additions provide a clearer mechanistic understanding of innate immune activation in the context of HIV infection.

  1. The connection of MPO with endothelial dysfunction is not clearly presented.

We thank the reviewer for highlighting the need to clarify the link between MPO and endothelial dysfunction. We have now added text in both the “Discussion” section and Figure 1 legend explaining how MPO-generated reactive species lead to LDL-C oxidation, foam cell formation, and endothelial injury. This establishes a direct mechanistic connection between MPO activity and vascular dysfunction in PLWH.

This is the added section within the discussion “Although MPO and markers of endothelial dysfunction (sICAM-1, sVCAM-1, sE-selectin) were often elevated, they were not consistently associated with clinical endpoints such as myocardial infarction or impaired FMD. However, MPO contributes to endothelial dysfunction primarily through its pro-oxidant activity, generating reactive species such as HOCl that oxidize LDL-C to form oxLDL-C [16–18]. Notably, oxLDL-C triggers vascular inflammation, recruitment of monocytes/macrophages, and foam cell formation, collectively impairing endothelial integrity and function [16–18]. Therefore, sustained MPO activation not only reflects oxidative stress but directly mediates vascular injury, linking chronic inflammation with early atherogenesis in PLWH on HAART [37; 41; 44]. Similarly, oxHDL-C was associated with higher BMI and lower apolipoprotein (Apo-AI), indicating altered lipid metabolism and potential CVD risk [40; 42]. These findings suggest that oxidative stress and lipoprotein modifications may be early contributors to endothelial dysfunction, independent of traditional lipid metrics.”

  1. The specific effects of different drug classes (e.g., NRTIs like abacavir, Protease Inhibitors, INSTIs) on MPO activity should be discussed, as the results suggest they may have divergent effects

We thank the reviewer for highlighting the need to discuss drug-class specific effects on MPO activity. We have now included a section in the “Discussion” clarifying that NRTIs such as abacavir may increase MPO-mediated oxidative stress, PIs are also associated with elevated oxidative markers, while NNRTIs and some INSTIs appear to have comparatively lower MPO activation. This emphasizes how different HAART regimens may differentially influence oxidative stress and cardiovascular risk in PLWH.

This is the newly added information “Notably, different drug classes appear to have divergent effects on MPO activity, for example, NRTIs like abacavir may increase MPO-mediated oxidative stress, PIs are also associated with elevated oxidative markers, whereas NNRTIs and some INSTIs may exert comparatively lower MPO activation. This suggests that the choice of HAART regimen could influence cardiovascular risk via differential impacts on oxidative stress and endothelial function [35; 39; 47; 48].”

Reviewer 3 Report

Comments and Suggestions for Authors

The paper is generally well-structured, following a logical flow from a general overview of MPO to its specific implications in PLWH and the influence of HAART. The systematic review methodology, as described, appears to adhere to established guidelines (PRISMA), which is commendable and enhances the credibility of the findings. The inclusion of a table summarizing the characteristics and key observations of the included studies is a valuable addition, providing a quick reference for readers.

Author Response

Reviewer 3

The paper is generally well-structured, following a logical flow from a general overview of MPO to its specific implications in PLWH and the influence of HAART. The systematic review methodology, as described, appears to adhere to established guidelines (PRISMA), which is commendable and enhances the credibility of the findings. The inclusion of a table summarizing the characteristics and key observations of the included studies is a valuable addition, providing a quick reference for readers.

We sincerely thank the reviewer for their positive and encouraging feedback. We appreciate the recognition of the manuscript’s structure, the systematic review methodology, and the value of the included summary table. Your comments affirm the clarity and rigor of our work, and we hope that the revised manuscript continues to meet the standards expected for publication.

Reviewer 4 Report

Comments and Suggestions for Authors

This is an interesting review article on a putative relationship between a sustained activation of enzyme myeloperoxidase in people suffering from human immunosufficiency virus and increased risk of inflammation-associated carrdiovascular disease in those patients. After the introduction, the Authors provide a general overview of myeloperoxidase and its pathological role, including discovery of myeloproteinase, the pathological link between myeloproteinase and endothelial dysfunction and pathological implications of myeloproteinase and human immunodefficiency.  The next section presents a systematic review evidence linking myeloproteinase activity with oxidative stress and inflammation-associated cardiovascular complications in immunodefficiency. This is followed by discussion study limitations and strengths and conclusions. The Authors concluded that despite the success of therapy there is still a need to further investigate the potential contribution of therapy regimen to the development of cardiovascular diseases. 

The article should be accepted after a small revision. Only two points should be taken in consideration:

1) Both the Figure 1 and the Table 1 look complicated and may contain too much infomation. The Authors should think how to make them easier to read especially by non-professional readrers.

2) The text should be supplied by a short summary containing a take-home message for readers.

I think this message may be a statement whether or not risk of cardiovascular disease is a side effect of the therapy, which requires furher study and a proper medical treatment. 

Author Response

Reviewer 4

This is an interesting review article on a putative relationship between a sustained activation of enzyme myeloperoxidase in people suffering from human immunosufficiency virus and increased risk of inflammation-associated carrdiovascular disease in those patients. After the introduction, the Authors provide a general overview of myeloperoxidase and its pathological role, including discovery of myeloproteinase, the pathological link between myeloproteinase and endothelial dysfunction and pathological implications of myeloproteinase and human immunodefficiency.  The next section presents a systematic review evidence linking myeloproteinase activity with oxidative stress and inflammation-associated cardiovascular complications in immunodefficiency. This is followed by discussion study limitations and strengths and conclusions. The Authors concluded that despite the success of therapy there is still a need to further investigate the potential contribution of therapy regimen to the development of cardiovascular diseases. 

The article should be accepted after a small revision. Only two points should be taken in consideration:

1) Both the Figure 1 and the Table 1 look complicated and may contain too much infomation. The Authors should think how to make them easier to read especially by non-professional readrers.

2) The text should be supplied by a short summary containing a take-home message for readers.

I think this message may be a statement whether or not risk of cardiovascular disease is a side effect of the therapy, which requires furher study and a proper medical treatment. 

We sincerely thank the reviewer for their thoughtful and constructive feedback. We appreciate the recognition of the manuscript’s structure, systematic approach, and relevance. In response to the specific suggestions, we have revised Figure 1 and Table 1 to improve clarity and readability (this is consistent with corrections from other reviewers of the manuscript), particularly for non-specialist readers, by simplifying visual elements and highlighting key points. Additionally, we have added a brief take-home message in the conclusion section emphasizing that while HAART effectively suppresses viral load, there remains a potential contribution of therapy regimens to cardiovascular risk in PLWH, which warrants further investigation and careful clinical management. All changes are clearly highlighted using track changes for ease of review.

The take home message “While antiretroviral therapy effectively controls HIV, certain regimens may contribute to cardiovascular risk through sustained oxidative stress and inflammation. Clinicians should remain vigilant, and further research is needed to optimize treatment strategies and minimize potential cardiovascular complications.”

Round 2

Reviewer 1 Report

Comments and Suggestions for Authors

The author's response to the reviewer comments is fair.

Reviewer 3 Report

Comments and Suggestions for Authors

No further comments, Congrats